# Effect of COVID-19 pandemic on missed medical appointment among adults with chronic disease conditions in Northwest Ethiopia

**Tadesse Awoke Ayele**[1], **Tesfa Sewunet Alamneh**[1], **Habtewold Shibru**[2], **Malede Mequanent Sisay**[1], **Tesfahun Melese Yilma**[3], **Melkitu Fentie Melak**[4], **Telake Azale Bisetegn**[5], **Tariku Belachew**[6], **Mahteme Haile**[6], **Taye Zeru**[7], **Mezgebu Selamsew Asres**[2], **Kegnie Shitu**[5]*

1 Epidemiology & Biostatistics Department, Institute of Public Health, College of Medicine and Health Sciences, University of Gondar, Gondar, Ethiopia, 2 Internal Medicine Department, College of Medicine and Health Sciences, University of Gondar, Gondar, Ethiopia, 3 Health Informatics Department, Institute of Public Health, College of Medicine and Health Sciences, University of Gondar, Gondar, Ethiopia, 4 Nutrition Department, College of Medicine and Health Sciences, University of Gondar, Gondar, Ethiopia, 5 Health Education & Behavioral Science Department, Institute of Public Health, College of Medicine and Health Sciences, University of Gondar, Gondar, Ethiopia, 6 Amhara Health Bureau, Bahir-Dar, Ethiopia, 7 Amhara Public Health Institute, Bahir-Dar, Ethiopia

* kegnsh@gmail.com

**Data Availability Statement:** The datasets used and/or analyzed during the current study are available as a supporting information.

## Abstract

### Background

COVID-19 had affected the health-care-seeking behavior of people with chronic medical conditions. The impact is even worse in resource-limited settings like Ethiopia. Therefore, this study was aimed to assess the extent and correlates of missed appointments among adults with chronic disease conditions before and during the COVID-19 pandemic in the Northwest Ethiopia.

### Methods

A retrospective chart review and cross-sectional survey were conducted from December 2020 to February 2021. A total of 1833 patients with common chronic disease were included by using a stratified systematic random sampling technique. Web-based data collection was done using Kobo collect. The data were explored using descriptive statistical techniques, the rate of missed appointments s before and during the COVID-19 pandemic was determined. A negative binomial regression model was fitted to identify the factors of missed appointment. An incidence rate ratio with its 95% confidence interval (CI) and p-value of the final model were reported.

### Results

The rate of missed appointments was 12.5% (95% CI: 11.13%, 14.20%) before the pandemic, increased to 26.8% (95% CI: 24.73%, 28.82%) during the pandemic (p-value <

**Funding:** This study was conducted by a fund received from the ministry of Health-Ethiopia. The funders had no role in study design, data collection and analysis, decision to publish, or preparation of the manuscript.

**Competing interests:** The authors have declared that no competing interests exist.

**Abbreviations:** NCDs, None Communicable Diseases; COVID-19, Novel Coronavirus; CVD, Cardiovascular Diseases; CLD, Chronic Liver Disease; CKD, Chronic Kidney Disease; IRR, Incidence Rate Ratio; IQR, Inter Quartile Range; COPD, Chronic Obstructive Pulmonary Disease; DM, Diabetes Mellitus.

0.001). Fear of COVID-19 infection and lack of transport was the most common reasons for missing appointments. Older patients (Adjusted Incidence Rate Ratio (AIRR) = 1.01, 95% CI: 1.001; 1.015), having treatment follow up more than 5 years (AIRR = 1.36, 95%CI: 1.103; 1.69), shorter frequency of follow-up (AIRR = 2.22, 95% CI: 1.63; 2.49), covering expense out of pocket (AIRR = 2.26, 95%CI: 1.41; 2.95), having a sedentary lifestyle (AIRR = 1.36, 95%CI: 1.12; 1.71), and history of missed appointments before COVID-19 pandemic (AIRR = 4.27, 95%CI: 3.35; 5.43) were positively associated with the incidence of missed appointments.

## Conclusion

The rate of missed appointment increased significantly during the COVID-19 pandemic. Older age, longer duration of follow up, more frequent follow-up, out-of-pocket expenditure for health service, history of poor follow-up, and sedentary lifestyle had positive relationship with missed appointments during the pandemic. Therefore, it is important to give special emphasis to individuals with these risk factors while designing and implementing policies and strategies for peoples with chronic diseases to ensure the continuity of care and to avoid the long-term impact on their health.

## Introduction

Novel Corona virus (COVID-19) has infected more than 221 million people and killed more than 4.5 million people globally as of September 2021And In Ethiopia, the number of infected and died reached 314,984 and 4,763, respectively [1]. The pandemic is still increasing and currently in its third wave, causing a restart of enforcement measures which might cause reduced access to health care in developing countries [2]. Access to health facilities was restricted due to fear of the spread of the virus from patient and health professional perspectives [3, 4]. Studies are deeply concerned about the impact of COVID-19 which have on patients suffering from chronic diseases, such as treatment delay, discontinuation, morbidity, and mortality [3, 5, 6].

Chronic diseases represent the leading cause of disability worldwide, with approximately one in three adults suffering from one or more chronic diseases [7]. Nearly 150 million patients with chronic diseases and 30 million with rare diseases continue to need care and treatment during these trying times as the services they rely on are pushed to capacity due to the current situation [8]. The COVID-19 pandemic is a more dangerous period for people with known chronic diseases including non-communicable diseases (NCDs) [9]. Although the virus potentially infects every individual, it is known that people with underlying chronic diseases have more severe symptoms. Studies show that among the increasing number of cases mostly affected populations are people with previously known chronic diseases [10]. Risk factors associated with serious disease and mortality are advanced age, cardiovascular disease (CVD), diabetes mellitus (DM), hypertension (HT), chronic lung disease, cancers, chronic kidney disease (CKD), use of immunosuppressive or biological agents, obesity, and smoking [11].

As of June 30, 2020, an estimated 41% of adults had delayed or avoided medical care including urgent or emergency care (12%) and routine care (32%) because of concerns about COVID-19 in the US [12]. About one-fifth of patients regardless of the disease state are

worried about maintaining their health during the corona virus outbreak. With all eyes on the corona virus, chronic diseases worsened, emergencies escalated, and countless care and treatments were delayed. The no-show rate was 17.2% during the COVID-19 pandemic [13]. A study in Singapore showed that 40% of patients with chronic medical conditions reported missing at least one healthcare appointment missing their health-care appointment during the COVID-19 outbreak [14]. A qualitative study in Belgium revealed disruption of the delivery of chronic care in the primary care context [15]. However, the rate of missed appointments in patients with common chronic diseases before and during the COVID-19 pandemic was not assessed in the study area. Moreover, as far as missed appoints are associated with poor health outcomes [16] and economic consequences [17], determining the magnitude and factors affecting it might have a paramount importance for designing appropriate strategies to enhance patients compliance for their treatment and improve their quality of life. Therefore, in this study, we assessed the rate of missed appointments and compared it between before and during COVID-19. Predictors of missed appointments during the pandemic were also identified using count data modeling.

## Methods

### Study design and period

A retrospective chart review and cross-sectional survey were conducted from December 2020 to February. 2021. Two years' retrospective data were collected; one year before the COVID-19 pandemic and one year during the COVID-19 pandemic.

### Study setting

The study was conducted in the Amhara regional state which is located in the northwestern part of Ethiopia. It has 15 Zones, and 180 Districts (139 rural and 41 urban). The region has 80 hospitals (6 referrals, 2 generals, and 72 primaries), 847 health centers, and 3,342 health posts. The study involved nine hospitals (referral, and district) which have chronic care and treatment centers.

### Source and study population

The source population was patients with common chronic diseases (HIV/AIDS, DM, CVD, Chronic Liver Disease (CLD), CKD, cancer, and Chronic Obstructive Pulmonary Disease (COPD) and on follow-up at the hospitals in Amhara regional state. Whereas, patients who had chronic care appointments and follow-up during the data collection period was the study population. All patients with common chronic diseases aged greater than or equal to 18 years who had been on medication for more than one year included before the pandemic were included. Those who were unable to communicate as well as patients with incomplete records were excluded.

### Sample size and sampling procedures

The sample size was calculated by using single population proportion formula by considering major objectives. Accordingly, the final sample size was 1833. All referral and selected district hospitals in the region were included to select study participants. First, stratification was done according to hospital status (referral or district). Later, hospitals were selected from each stratum. Finally, the study participants were selected using a systematic random sampling technique with disease types (HIV/AIDS, DM, CVD, COPD, CLD, CKD, and cancer) in the

selected hospitals while considering the proportion of disease categories. The total sample was distributed across diseases types proportionally.

## Data collection methods and measurements

Primary and secondary sources were used to collect the desired data on the variables of interest. Two years of retrospective data were extracted from patients' charts. Charts were retrieved from the treatment centers in the selected hospitals. Data collection was conducted approximately a year into the pandemic. One-year retrospective data were extracted for the same patients before the COVID-19 pandemic. Frequency of missed appointment which was used as an outcome variable and all other available epidemiological information was collected including, clinical, demographics variables, risk factors, exposures, lab results, and patient outcomes. Health management information systems and patient charts were used to extract the data using chart abstraction form. We extracted two-year's data from medical chart (the primary data source). For those variables that were not recorded in the chart, we conducted patient interview to collect the data (the secondary data source). Patients' interviews were made after the appointment logbook and patient chart was retrieval.

## Operational definition

**Chronic diseases.** Chronic diseases are broadly defined as diseases that last 1 year or more and require ongoing medical attention or limited activities of daily living or both [18]. In our setup, chronic diseases under organized follow-up are cardiovascular diseases (hypertension, CKD, and cardiac illnesses), cancer, Respiratory disorder (COPD and Asthma), diabetes, chronic liver diseases, and HIV/AIDS [19].

**Missed appointment.** When a patient did not attend the follow-up according to the physicians' schedule.

## Quality assurance mechanism

To maintain the quality of the data, training was given to data collectors and supervisors. Trained nurses and medical doctors who know the treatment centers were recruited. The questionnaire was translated to the local language Amharic and back to the English version to assure its consistency. Web-based data collection was used and the questionnaire was changed into electronic form using Kbotoolbox. A pretest was carried out and possible amendments and an internal consistency reliability test were done. The collected data were reviewed on daily basis centrally for completeness and consistency.

## Data management and analysis

Following completion of data collection, the web-based data was exported to STATA and R for management and analysis. Cleaning, coding, categorization and error inception were made by the research team. Results were explored using descriptive statistical techniques and prevalence, mean, median, Inter Quartile Range (IQR), Standard deviations, a test of associations were computed. Subsequently, proportion with its 95% CI was calculated to assess the difference in proportion of missed appointments s between before and during the COVID-19 pandemic. In addition, Poisson regression with an offset variable of duration of follow up was considered to identify the predictors for frequency of missed appointment. The Poisson regression has constrained assumption that is the mean and the variance should be equal which was tested by using Vuong test [20]. Finally, negative binomial regression was fitted to identify determinants of missed appointments with over dispersion during the COVID-19

pandemic [21]. A 95% confidence interval, Incidence Rate Ratio (IRR), and p-value of the final model were reported to determine statistical significance.

## Ethics approval and consent to participate

Ethical clearance was secured from the institutional review board of the University of Gondar. The supportive letter was obtained from the Amhara public health institute and permission was obtained from the medical director of each hospital. Participants of the study were informed about the purpose, objectives, and their right to participate or not participate in the research. The right of participants to withdraw from the study at any time, without any pre-condition was disclosed unequivocally. Written consent was obtained from each participant before data collection. Moreover, to guarantee confidentiality code numbers were used rather than personal identifiers.

## Results

### Socio-demographic characteristics

A total of 1,833 individuals with common chronic diseases were included with a response rate of 99%. Accordingly, 1815 patients were included in the study from whom 1005 (55.37%) were female. The median age of the study participants was 48 years with an IQR of 22 years (37–59). The majority (83.92%) of the study participants were orthodox followers and 1,262 (69.53%) were urban dwellers. The median monthly income of the participants was 2200 ETB with an IQR of 3000 ETB (1000 ETB—4000 ETB). More than one-fourth (28.0%) of the individuals with a common chronic condition were not able to read and write (Table 1).

Regarding access to health facilities, the median distance to the health facility was 7 KM with an IQR of 31 KM (3–34). More than half of the patients (55.67%) live within 10 kilometers distance from the nearby hospital, while about 5% of the patients live more than 100 kilometers away from the nearby hospital. The living arrangement of nearly half (46%) of the patients was mothers of whom 175 (21.1%) were the head of their family. Among the patients, 661 (36.46%) were head of their household and nearly one-in-three (33.65%) were fathers. Most of the patients (43%) use health insurance and followed by out-of-pocket payments accounting for 42% for the service provided during their follow-up. Only 3% use poverty cards to cover their service cost (**Table 1**).

The three common types of diagnosis were HIV/AIDS (27.69%), hypertension (23.41%), and diabetic Mellitus (22.7%). About one in four patients (23.63%) have been receiving follow-up for more than 10 years with a median duration of 6 years (IQR = 3–10 years). Nearly one in five (20.13%) patients had an emergency appointment or admission during the COVID-19 pandemic period. Most (74.6%) have had unusual sudden onset of COVID-19 like symptoms during the pandemic period. Dry cough (22.25%), headache (17.0%), shortness of breath (13.64%), and high-grade fever (11.76%) were the common symptoms experienced by the patients. However, only 33 (1.8%) were tested positive for COVID-19. Although a significant number of patients experienced COVID-19 symptoms, most 974(54.7%) perceived that they were never infected with the virus.

Nearly one-third (33.35%) of the patients had identified co-morbid diseases in which hypertension, 252 (41.9%), was the commonest type of co-morbidity (**Table 2**).

**Missed appointment.** About 225 (12.47% (95% CI: 11.13%, 14.20%)) of patients with common chronic diseases missed their appointment during their follow-up before COVID 19 pandemic. However, the missed appointment was increased by more than double during the COVID-19 pandemic (26.8% (95% CI: 24.73%, 28.82%) (Fig 1). The change was statistically significant (p-value < 0.001). Moreover, the two confidence interval estimatesdid not over-

**Table 1. Background characteristics of patients with common chronic diseases in Amhara region, Ethiopia, 2021.**

| Variables | Category | number | % |
|---|---|---|---|
| Sex | Female | 1,005 | 55.37 |
| | Male | 810 | 44.63 |
| Religion | Orthodox | 1,519 | 83.92 |
| | Muslim | 276 | 15.25 |
| | Protestant | 2 | 0.11 |
| | *Others | 5 | 0.28 |
| Residence | Urban | 1,262 | 69.53 |
| | Rural | 553 | 30.47 |
| Marital status | Married | 1,187 | 65.40 |
| | Single | 218 | 12.01 |
| | Divorced | 107 | 5.90 |
| | Separated | 115 | 6.34 |
| | Widowed | 188 | 10.36 |
| Educational status | Unable to read and write | 508 | 27.99 |
| | Able to read and write | 293 | 16.14 |
| | Primary education | 326 | 17.96 |
| | Secondary education | 298 | 16.42 |
| | Diploma | 206 | 11.35 |
| | Degree and above | 184 | 10.14 |
| Occupation | Government employee | 411 | 22.55 |
| | Private employee | 165 | 9.05 |
| | Farmer | 323 | 17.72 |
| | Merchant | 195 | 10.70 |
| | Unemployed | 145 | 7.95 |
| | Housewife | 429 | 23.53 |
| | Student | 88 | 4.83 |
| | **other | 67 | 3.68 |
| Distance from hospital | <10km | 1,012 | 55.67 |
| | 11–50 km | 540 | 29.7 |
| | 51–100 km | 183 | 10.07 |
| | >100 km | 83 | 4.57 |
| Living arrangement | Head of the family | 661 | 36.46 |
| | Mother | 834 | 46.00 |
| | Father | 610 | 33.65 |
| | Son/Daughter | 185 | 10.20 |
| | Relative | 65 | 3.59 |
| | Other | 24 | 1.32 |
| Payment method | Health Insurance | 784 | 42.98 |
| | Poverty card | 60 | 3.29 |
| | Out of Pocket | 775 | 42.49 |
| | Waived | 205 | 11.24 |

*Protestant and catholic

**daily laborer, driver, retired, clergyman, broker, promoter, soldier, etc

lapped indicating a significant difference. Fear of COVID-19 infection 308 (64.2%) and lack of transport 148 (30.83%) were the most common reasons mentioned for missing their appointment.

**Table 2. Clinical factors of the study participants.**

| Variables | Category | number | percent |
|---|---|---|---|
| Duration of follow up | < 2years | 373 | 20.45 |
| | 2–5 years | 534 | 29.28 |
| | 6–10 years | 486 | 26.64 |
| | >10 years | 431 | 23.63 |
| Emergency appointments during COVID 19 | Yes | 367 | 20.13 |
| | No | 1,456 | 79.87 |
| COVID 19 Symptoms | Dry cough | 403 | 22.25 |
| | High grade fever | 213 | 11.76 |
| | Loss smell | 58 | 3.20 |
| | Muscle/joint ache | 122 | 6.74 |
| | Headache | 308 | 17.01 |
| | Shortness of breath | 247 | 13.64 |
| | No symptom | 1,246 | 68.76 |
| Perceived COVID-19 infection | Most likely | 121 | 6.8 |
| | Likely | 105 | 5.9 |
| | Less likely | 111 | 6.24 |
| | Can't decide | 469 | 26.35 |
| | Never | 974 | 54.72 |
| Type of medication | Oral | 1,589 | 87.16 |
| | Injectable | 335 | 18.38 |
| | Both | 128 | 7.02 |
| Frequency of follow up | Weekly | 25 | 1.37 |
| | Every two/three weeks | 91 | 5.00 |
| | Monthly | 554 | 30.42 |
| | Every two month | 402 | 22.08 |
| | Every 3–5 months | 585 | 32.13 |
| | Every 6 or more months | 164 | 9.01 |
| Reasons for missed appointment | Fear of COVID 19 infection | 308 | 64.17 |
| | Health facility lockdown | 61 | 12.71 |
| | Health professional were not cooperative | 12 | 2.50 |
| | Transportation problem | 148 | 30.83 |
| | *Others | 80 | 1.67 |
| Presence of co-morbidity | Yes | 602 | 33.35 |
| | No | 1,203 | 66.65 |

Missed appointments before and during the COVID-19 pandemic varied with the type of chronic diseases. Patients with cardiovascular disorders had shown a significant increment from10.08% before COVID 19 pandemic to 31.69% during the pandemic. However, a small increment was observed among HIV/AIDS patients that increased from 17.8% before COVID 19 pandemic to 19.6% during the Pandemic (Fig 2).

**Factors associated with the frequency of missed appointments.** Poisson regression was used to model the number of missed appointments over the demographic, clinical, and behavioral factors. Besides, the assumption of the equality of mean and variance was checked using Vuong test. The evidence suggested that the violation the equality of mean and variance the assumption and indicated that the presence of over dispersion. Consequently, the model was extended to negative binomial regression. Similarly, the presence of excess zero was checked and zero-inflated Poisson regression was also fitted. The three models were compared and

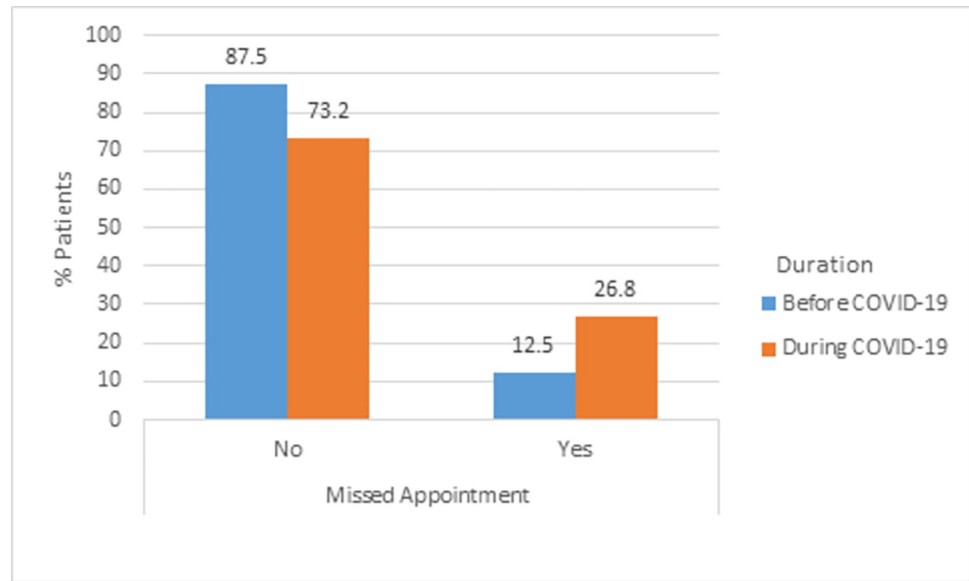

**Fig 1. Missed appointment before COVID-19 pandemics and during the pandemics among common chronic disease patients in Amhara Region.**

negative binomial regression was found to be better than the other. Accordingly, the variables age, religion, marital status, education status, distance from the hospital, duration on follow up, frequency of follow up, payment method, COVID-19 symptom, sedentary lifestyle, and missing appointments before COVId-19 pandemic were found to be significant predictors in the bi-variable analysis at 0.2 level of significance. Finally, age, religion, duration of follow-up, frequency of follow-up, payment methods, COVID-19 symptoms, sedentary lifestyle, and history of missing appointments before the COVId-19 pandemic were statistically significant in the multivariable analysis.

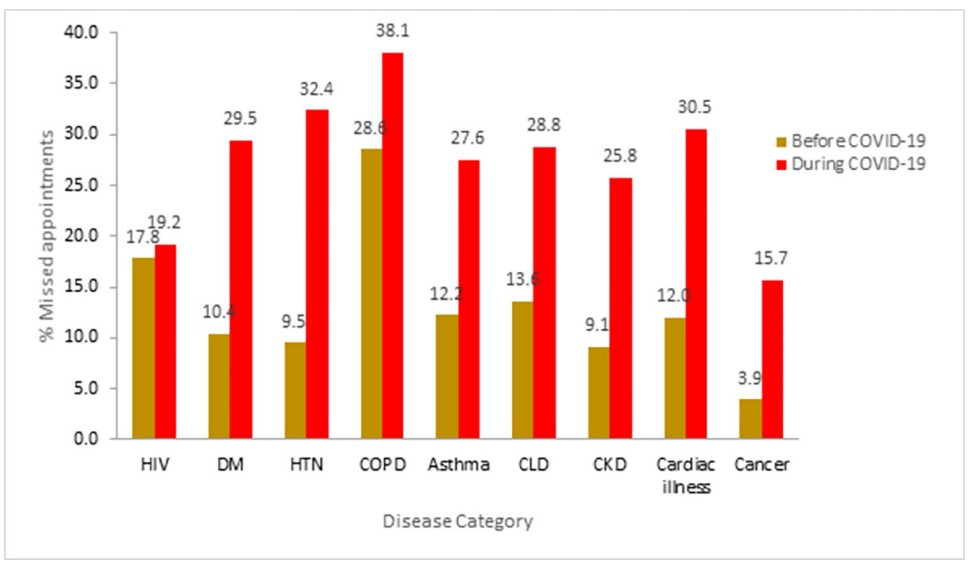

**Fig 2. Missed appointment before COVID 19 pandemic and during the pandemic among common chronic disease patients in Amhara Region.**

**Table 3. Negative binomial regression for missed appointments over demographic, clinical, and behavioral factors.**

| Variable | Category | Crude Incidence rate ratio (CIRR) (95% CI) | AIRR (95% CI) | P-value |
|---|---|---|---|---|
| Age (years) | | 1.43(1.13–1.81) | 1.01 (1.001; 1.015) | 0.025 |
| Religion | Christian | 1 | 1 | |
| | Muslim | 1.488(1.13–1.97) | 1.70 (1.33; 2.85) | <0.0001 |
| Marital status | Married | 1 | 1 | |
| | Single | 0.82 (0.61, 1.20) | 1.12 (0.77; 1.63) | 0.365 |
| | Divorced | 0.74 (0.52, 1.04) | 0.99 (0.71; 1.40) | 0.918 |
| | Widowed | 0.83 (0.58, 1.18) | 0.91 (0.64; 1.29) | 0.657 |
| Education Status | No Formal Educ. | 1.73 (1.12; 1.70) | 1.17 (0.92; 1.48) | 0.293 |
| | Formal Educ. | 1 | 1 | |
| Distance from Hospital | < = 30 KM | 1 | 1 | |
| | >30 KM | 1.23 (0.97; 1.56) | 1.15 (0.90; 1.47) | 0.228 |
| Duration on follow-up | < 6 Years | 1 | 1 | |
| | >5 Years | 1.21(0.98; 1.50) | 1.25 (1.103; 1.69) | 0.032 |
| Frequency of follow up | < = 30 days | 2.22(1.80; 2.73) | 2.09 (1.63; 2.49) | <0.0001 |
| | >30 days | 1 | 1 | |
| Payment Method | Out of Pocket | 2.26 (1.57; 3.26) | 2.04 (1.41; 2.95) | <0.0001 |
| | Health Insurance | 2.85 (1.97; 4.09) | 2.09 (1.41; 3.09) | 0.001 |
| | Waived | 1 | 1 | |
| COVID-19 symptom | Yes | 1 | 1 | |
| | No | 1.27 (1.01; 1.60) | 1.25 (1.01; 1.57) | 0.039 |
| Current alcohol use | Yes | 1.51(1.14; 2.01) | 1.43 (1.09; 1.88) | 0.063 |
| | No | 1 | 1 | |
| Sedentary behavior | No | 1 | 1 | |
| | Yes | 1.69(1.36; 2.09) | 1.39 (1.12; 1.71) | 0.003 |
| Missed appointments before COVID | Yes | 3.895(3.01; 5.04) | 4.27 (3.35; 5.43) | <0.0001 |
| | No | 1 | 1 | |

A year increase in age of the patient increases the incidence of missed appointments by 1% (AIRR = 1.01, 95%CI: 1.001; 1.015). Patients who followed treatment for more than 5 years had a 36% (AIRR = 1.36, 95%CI: 1.103; 1.69) increase in the incidence of missed appointments. The more frequent the visits, the higher the incidence of missed appointments. The incidence of missed appointments increased two-fold (AIRR = 2.22, 95%CI: 1.63; 2.49) when the frequency of follow-up is less than a month as compared to greater than a month.

The incidence of missed appointments was about two-fold (AIRR = 2.26, 95% CI: 1.41; 2.95) for those patients who covered their medical expenses out of their pocket as compared to those who were waived. Those who never experienced any of the COVID-19 symptoms were at higher risk of missing their appointments as compared to those who experienced at least one symptom. The incidence of missed increased by nearly 40% for patients who had a sedentary lifestyle (AIRR = 1.36, 95%CI: 1.12; 1.71) and a four-fold increase in the incidence of missed appointments for patients who had missed their appointments before the COVID-19 pandemic (AIRR = 4.27, 95% CI: 3.35; 5.43) (**Table 3**).

## Discussion

The emergence of the COVID-19 pandemic in December 2019 becomes the greatest challenge in the world after World War II with catastrophic global impact [22]. Still, after a year, the pandemic remains very challenging, especially for low-income countries. Ethiopia is also one of the victims for this global challenge. There have been different strategies wearing, frequent had

washing, and other non-pharmaceutical interventions implemented to avert this epidemic, however, the implementation level have been very limited due to the cultural and traditional practices in the country. Though the pandemic affected every segment of the population, the impact has been paramount in patients with common chronic diseases [6]. In this multicenter study, we reviewed the medical record and interviewed 1815 adults with common chronic diseases to assess the missed appointments before and during the COVID-19 pandemic and its associated factors in Northwest, Ethiopia. The rate of missed appointments was significantly higher during the COVID-19 pandemic period as compared to before the pandemic. It was increased from 12.47% to 26.8%. This finding was in line with findings reported by previous studies: claimed the magnitude of missed medical appointments increased following the introduction to the pandemic. The possible reason may be because patients may fear contracting the novel corona virus if they go out of their house and visit health facilities. This was supported by the findings of the present and the previous studies [12, 14, 23]: the main reason for missing medical appointments was fear of COVID-19 infection. However, the magnitude of missed medical appointments claimed by this study was lower than what has been reported in the United States [12], Singapore [14], Canada [24], and Italy [25], where 41%, 40%, 38% and 77.4% of adults with chronic medical patients missed their medical appointments during the pandemic respectively. This discrepancy may be attributed to the difference in the transmission rate of COVID-19 among study areas [1]: the rate of COVID-19 transmission is higher in United States, Singapore, Canada and Italy compared to the rate of COVID-19 transmission that has been reported in Ethiopia. Given this, patients who live in areas with a higher COVID-19 transmission rate might be more likely to miss their medical appointments than patients who live in areas with a lower COVID-19 transmission rate. Fear of COVID-19 infection 308 (64.2%) and lack of transport 148 (30.83%) were the most common reasons for missing their appointments. Similar to our finding, a narrative review showed that in-person care for individuals with chronic diseases has decreased due to government restriction of elective and non-urgent healthcare appointments, and greater instilled fear over potential COVID-19 exposure during in-person appointments (21). This is further supported by the fact that many people fear spending time in a waiting area exposed to other patients, increasing their risk of exposure to COVID-19. Particularity, it may be a commonly observed phenomenon in developing countries where long time stay at waiting areas/room in the health facilities is highly expected.

Multivariable analysis of the present study revealed that, older age,—duration of the illness, frequency of follow up, out of pocket expenditure for health service, history of missed medical appointments before COVID-19, experienced COVID-19 like symptom(s), and sedentary lifestyle were found to be factors associated with missed medical appointments among adults with chronic medical conditions.

In this study, the likelihood of missing medical appointments was more likely among patients with older age. This finding was inconsistent with previous studies conducted elsewhere [14, 26, 27]. On the other hand, the finding was supported by previous studies: claimed that older age increases the likelihood of canceling medical appointments [28–30]. This may be because older patients have a higher risk of developing a severe form of COVID-19 compared to younger patients. Due to this, older patients may fear contracting COVID-19 a lot and cancel their medical appointments. This finding implied the need to give due attention to establish tracing mechanisms to reduce missing medical appointments among older adults with chronic medical conditions since they are at higher risk of developing complicated cases and death from their chronic medical condition(s) than younger ones with chronic medical conditions [31]. Moreover, older patients may be dependent on family to take them to their appointment and tend to miss their hospital appointments if families are unavailable to do so.

Patients with a longer duration of the illness were more likely to miss their medical appointments. This may be due to that patients who lived a long time with the disease condition may have good self-care of their illness and therefore they may have better confidence in managing their illness by themselves [29], Therefore, they may cancel their medical appointments more likely to patients with shorter duration of the illness.

The current analysis revealed that patients who have a shorter frequency of follow-up were more likely to miss their medical appointments. This finding was inconsistent with findings reported by studies done in Italy [27] and Mexico [32], where a shorter frequency of follow-up was assorted with the likelihood of missing medical appointments. This may be because the perceived threat of contracting COVID-19 may be higher among chronic patients with shorter follow up since they are expected to visit health facilities more frequently than in turn increase their risk of exposure to the pandemic. Thus, they would prefer to cancel their medical appointment to reduce their risk of COVID-19 infection.

The incidence of missed medical appointments was four-fold higher among patients who had a history of missed appointments before the COVID-19 pandemic. This finding was consistent with findings reported by previous studies: patients who had missed previous appointments were more likely to miss their future visit [13, 14, 26, 27]. This may be because missed appointments might become habitual [24]. Moreover, these finding highlights the need to intervene with behavior change interventions among patients who have a history of missed medical appointments to overcome further development of the habit of missing medical appoints.

The incidence of missed appointments was about two-fold for those patients who covered their medical expenses out of their pocket as compared to those who were waived. This finding was supported by a previous study conducted in Oman [33]. This may be due to that patients may have no health coverage to pay for their treatment or they have limited coverage that does not cover every health service. On the other hand, it can be also explained by the fact that since they pay for the health service, they might prefer to go to a private hospital without having to wait for their appointment since in both scenarios they will end up paying for their treatment unlike to patients who received waived health services only form public health facilities. Furthermore, the pandemic has increased that financial strain by causing large-scale unemployment, loss of insurance, and a shift in priorities. Concerning healthcare costs, providers are encouraged to work with patients who have a financial hardship caused or affected by the pandemic. Helping patients during this difficult time by making alternative payment arrangements, such as payment plans, will hopefully increase patient safety, decrease potential claims, and establish a long-term patient's adherence to their medical appointments that will extend beyond the COVID-19 pandemic.

Those who never experienced any of the COVID-19 symptoms were at higher risk of missing their appointments as compared to those who experienced at least one symptom. This could be because patients who experienced COVID-19 may be more likely to visit health facilities for confirmation and therefore it may reduce their chance to cancel their medical appointments.

There is nearly 40% increase in the incidence of missed appointments for patients who had sedentary lifestyles. This could be explained by patients who are engaged in sedentary lifestyles may be low health literacy and are less likely to understand adequately what they are expected to do with their chronic condition. In this point of view, they may be more likely to miss their medical appointment compared to patients who are not engaged in sedentary lifestyles.

This study has several strengths and limitations. The strength of the study includes the multicenter nature of the study, the large sample size which has impact on the power of the study, and data were collected from multiple sources which increase the generalizability of our

findings. Moreover, the respondents were also informed about the importance of the study and the confidentiality of personal data to gain the trust of respondents and minimize the non-response rate. But this study was not free of limitations. Since the study was facility-based there might be a risk of social desirability bias. Another limitation is the retrospective nature of the study. When the date for an appointment was not documented, we could not determine whether an individual actually missed the appointment or not. Moreover, this study considered both cancelled and rescheduled appointments as missed appointments and might have overestimated the number of missed appointments.

## Conclusion

The rate of missed appointments in patients with a chronic medical condition had shown significant increment during the COVID-19 pandemic. The rate increased to more than double during the COVID-19 pandemic period. Chronic disease patients who missed their appointments before the pandemic were more likely to miss their appointments during the pandemic. Fear of COVID-19 infection and lack of transport were the main reasons. Although most of the chronic medical condition patients had at least one onset of the symptoms of COVID only a few of them were tested. Longer duration follow-up time, shorter frequency of follow-up, covering expense out of pocket, alcohol drinking, not experienced COVID-19 like symptoms, and had a sedentary lifestyle increased the incidence of the missed appointment. There is an urgent need for better chronic disease management strategies to avoid missing appointments. Every chronic condition patient should be tested for COVID 19 whenever they experience the symptom. In addition, the ultimate goal of chronic disease follow up is to control and slow the disease progression, and prevention of related complication. Missing medical appoint had a significanct effect on these chonic diease follow up and mangment goal. Therfore, it is better to give special emphasis to individuals with older age, longer duration of follow up, frequent follow-up, cover health expenditures from their pocket, history of missed appointment, sedentary lifestyle while designing and implementing policies and strategies for peoples with chronic diseases to ensure the continuity of care and to avoid the long-term impact on their health.

## Supporting information

**S1 Text. Survey tool of missed medical appointment among adults with chronic disease conditions in Northwest Ethiopia.**
(DOCX)

**S1 Dataset. De-identified data of missed medical appointment among adults with chronic disease conditions in Northwest Ethiopia.**
(XLSX)

## Acknowledgments

We would like to thank the University of Gondar, Amhara Health Bureau, and Amhara Public Health Institute for the technical support and facilitation they provide during the study. We also thank the study participants for providing the information during the interview.

## Author Contributions

**Conceptualization:** Tadesse Awoke Ayele, Tesfa Sewunet Alamneh, Habtewold Shibru, Malede Mequanent Sisay, Tesfahun Melese Yilma, Melkitu Fentie Melak, Telake Azale Bisetegn, Tariku Belachew, Mahteme Haile, Taye Zeru, Kegnie Shitu.

**Data curation:** Tadesse Awoke Ayele, Tesfa Sewunet Alamneh, Tesfahun Melese Yilma, Mezgebu Selamsew Asres, Kegnie Shitu.

**Formal analysis:** Tadesse Awoke Ayele, Tesfa Sewunet Alamneh, Tesfahun Melese Yilma, Kegnie Shitu.

**Funding acquisition:** Tadesse Awoke Ayele.

**Investigation:** Tadesse Awoke Ayele, Tesfa Sewunet Alamneh, Malede Mequanent Sisay, Tesfahun Melese Yilma, Melkitu Fentie Melak, Taye Zeru, Kegnie Shitu.

**Methodology:** Tadesse Awoke Ayele, Tesfa Sewunet Alamneh, Habtewold Shibru, Malede Mequanent Sisay, Tesfahun Melese Yilma, Melkitu Fentie Melak, Telake Azale Bisetegn, Kegnie Shitu.

**Project administration:** Tadesse Awoke Ayele, Tesfa Sewunet Alamneh, Malede Mequanent Sisay, Tesfahun Melese Yilma, Melkitu Fentie Melak, Mahteme Haile, Taye Zeru, Kegnie Shitu.

**Resources:** Tadesse Awoke Ayele, Tesfa Sewunet Alamneh, Habtewold Shibru, Tesfahun Melese Yilma, Kegnie Shitu.

**Software:** Tadesse Awoke Ayele, Tesfahun Melese Yilma, Kegnie Shitu.

**Supervision:** Tadesse Awoke Ayele, Tesfa Sewunet Alamneh, Habtewold Shibru, Malede Mequanent Sisay, Melkitu Fentie Melak, Tariku Belachew, Taye Zeru, Kegnie Shitu.

**Validation:** Tadesse Awoke Ayele, Tesfa Sewunet Alamneh, Tesfahun Melese Yilma, Melkitu Fentie Melak, Mezgebu Selamsew Asres, Kegnie Shitu.

**Visualization:** Tadesse Awoke Ayele, Tesfa Sewunet Alamneh, Tesfahun Melese Yilma, Mezgebu Selamsew Asres, Kegnie Shitu.

**Writing – original draft:** Tadesse Awoke Ayele.

**Writing – review & editing:** Tadesse Awoke Ayele, Tesfa Sewunet Alamneh, Habtewold Shibru, Malede Mequanent Sisay, Tesfahun Melese Yilma, Melkitu Fentie Melak, Telake Azale Bisetegn, Tariku Belachew, Mahteme Haile, Taye Zeru, Mezgebu Selamsew Asres, Kegnie Shitu.

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
