## [Decision Letter · Decision Letter 0]

17 Mar 2022

PONE-D-21-28858Effect of COVID-19 pandemic on missed medical appointments among adults with chronic diseases conditions in Northwest, EthiopiaPLOS ONE

Dear Dr. Shitu,

Thank you for submitting your manuscript to PLOS ONE. After careful consideration, we feel that it has merit but does not fully meet PLOS ONE’s publication criteria as it currently stands. Therefore, we invite you to submit a revised version of the manuscript that addresses the points raised during the review process.

The reviewers have raised a number of concerns in their comments; please ensure that you address each of the reviewers' comments in revising your manuscript. Please also note the technical requests below regarding 1) participant consent, and 2) the survey or questionnaire used in this study.

We look forward to receiving your revised manuscript.

Kind regards,

Hugh Cowley

Senior Editor

PLOS ONE

Journal Requirements:

Reviewers' comments:

Reviewer's Responses to Questions

**Comments to the Author**

1. Is the manuscript technically sound, and do the data support the conclusions?

Reviewer #1: Yes

Reviewer #2: Partly

2. Has the statistical analysis been performed appropriately and rigorously? 

Reviewer #1: Yes

Reviewer #2: No

3. Have the authors made all data underlying the findings in their manuscript fully available?

Reviewer #1: Yes

Reviewer #2: No

4. Is the manuscript presented in an intelligible fashion and written in standard English?

Reviewer #1: No

Reviewer #2: No

5. Review Comments to the Author

Reviewer #1: Thank you for the opportunity to review this paper, which looks at pre and during Covid-19 rates of missed appointments in Ethiopia. It was very interesting to read, and is very topical and provides new and useful information on the topic, and how Covid-19 has impacted on how many appointments are missed- which is a very relevant and potentially useful area for policy and practice.

I have made a few comments, that will help to increase the readability of the paper, and some suggested additions to the discussion. There are a few instances of language needing a bit of attention (words missing or just wrong tense), and I have highlighted a few but would suggest authors have another check through.

Abstract:

Results: There seems to be a missing word where it reads ‘most (55.4%) were female.’ Suggest need to add in ‘participants’ or similar.

Conclusion: This is repetitive as it duplicates the reasons of missed appointments which is also found in results. The conclusions would be better to include more about the implications of the findings or the next steps for future research. This relates to a later comment about the conclusion in the main body of the paper.

Introduction:

The first paragraph of the introduction would benefit from more references. If the references that are provided also cover earlier points, it might be worth adding the reference number in again. Some points seem to be unreferenced so this would make it clearer. One example of this is ‘Access to health facilities was restricted due to fear of the spread of the virus from patient and health professional perspectives. Patients were scared to go to a health facility to receive their care and treatments.’ This occurs throughout the introduction.

In the second paragraph of the introduction you state that ‘Although the virus infects every individual’ this would read better as ‘potentially infects every individual’ as much is still unknown and not everyone is infected.

When you say ‘An estimated 41% of adults had delayed or avoided medical care including urgent or

emergency care (12%) and routine care (32%) because of concerns about COVID-19 in the US’ it would be helpful to define the timeframe that this covers.

It would be useful to include a statement on why looking at missed appointments is needed/ relevant/ interesting. What is the impact of missed appointments on health services and patients? (cost and time for health care professionals and unmet need/ delayed treatment for patients).

Methods:

Study design and period: ‘Two years retrospective data’ could you just briefly state what type of data this was- e.g. was it electronic patient records etc?

Study setting: Of the 9 hospitals that participated in the study, what were their demographics? (e.g were they rural/ urban etc).

Data collection methods and measurement:

This sentence would be worth rewording to make it clearer: ‘During data collection, it has been about a year since the emergence of the COVID-19 pandemic.’ Something like ‘Data collection was conducted approximately a year into the pandemic.’

On page 7 you mention patient interviews. This is the first mention, and it is not clear what this relates to. Is this a reference to the surveys that were used? It implies that there is a qualitative component to the study, so suggest rewording it.

Results:

Socio-demographic characteristics: In line with earlier comment, when you say ‘A total of 1,833 individuals with common chronic diseases were interviewed’ do you mean ‘A total of 1,833 individuals with common chronic diseases completed the survey.’ Possibly reword to make it clearer they completed the survey.

‘More than one-fourth (28.0%) of the individuals with a common chronic condition was not able to read and write’ should say ‘were’ not ‘was.’

There are a few examples in the results sections where the language would benefit from being revised to make it clearer. E.g. ‘About one in four patients (23.63%) have been following for more

than 10 years,’ (have been receiving follow-up?) and ‘However, only 33 (1.8%) diagnoses for COVID-19 and all of them were positive,’ (Diagnosed implies that they were positive) ‘Nearly one-third (33.35%) of the patients had identified co-morbid diseases in which hypertension was the commonest type of co-morbidity observed in most of the patients.’ (duplication- commonest and most- suggest just use one or the other).

Missed appointments: Again a few suggestions on improving the language. E.g. ‘About 225 (12.47% (95% CI: 11.13%, 14.20%)) of patients with common chronic diseases have missed their appointment during their follow-up before COVID 19 pandemic’ (remove ‘have’).

Discussion:

You start by discussing generally about Covid and low-income countries, but then move on to talk specifically about the country being studied. ‘There have been different strategies and non-pharmaceutical interventions implemented, however, the implementation level has been very limited due to the cultural and traditional practices in the country’. It would be good to state here when you move on to talk about one specific country to make it clear.

Again a few missed words in this section ‘the rate of COVID-19 transmission is higher United States, Singapore, Canada and Italy.’ Missing ‘in’.

Strengths and limitations: There is limited discussion of the strengths and limitations for this study. You mention a few strengths but no discussion of why they are strengths or how they strengthen the study.

I also suggest you could include a section on implications and discuss how this research can help understanding of this topic, implementation of policy or impact on practice. I also think that it would be useful to include any future research needed or [planned on the topic.

This also needs to be included in the abstract.

Reviewer #2: The authors conducted a retrospective cohort study to evaluate the impact of the COVID-19 pandemic on missed appointments among those with chronic conditions in Northwest of Ethiopia. It was important to know the scale of the impact and the risk factors that were associated with increased risk of no-show. Here are my concerns:

1. I do not know why the author called their study “chart review and cross-sectional survey”. Their data source was electronic chart. I also did not see any conventional survey was conducted. Suggest remove “chart review and cross-sectional survey” and replace it with “a retrospective cohort study”.

2. Appointment data were extracted during the COVID-19 pandemic and one year before the pandemic for the same patients. Thus, the outcome, missed appointments from these two periods were not independent. However, the authors used an independent t-test to test the difference in missed appointments before and during the pandemic. In addition, t-test may not be the right test for the number of missed appointments. A binomial test is preferred for count data, or just run the negative binomial regression model without including covariates.

3. In the Results, the authors described how they landed on negative binomial regression because of overdispersion. I recommend a) In Methods, make clear that number of missed appointment is the dependent variable in the negative binomial regression; b) consider the dependence of observations in the regression; c) consider including number of scheduled appointments as an offset in the regression model (natural log(number of scheduled appointments)); when the offset is used, they should remove “The more frequent the visits, the higher the incidence of missed appointments” in page 14, and should not include duration of follow-up because number of scheduled appointments already contained all the information needed.

4. It was not clear what criteria were used to determine that negative binomial regression model was the best. Please provide a reference in page 14.

5. These sentences in Discussion are confusing: “Still, after a year, the pandemic remains very challenging, especially for low-income countries. There have been different strategies and non-pharmaceutical interventions implemented, however, the implementation level has been very limited due to the cultural and traditional practices in the country.” We are in the third year of the pandemic. I do not know what “strategies” and “interventions” were referred to and for what purposes. Suggest removing these sentences.

6. The author did not distinguish missed appointments, cancelled appointments, and rescheduled appointments. Were cancelled and rescheduled appointments counted as missed appointments? If they were, how many of those missed appointments were cancelled or rescheduled? Please discuss this issue.

7. There were many grammar-like errors in the manuscript, some examples are: a) lower case for “No-show rate” in page 5; b) there should be a space between 1005 and (55.37%) in page 8; c) throughout the manuscript, insert a space in 95%CI (like 95% CI); d) need define abbreviations before using them (e.g., CIRR, AIRR); e) decimals were not consistence, for example, in Table 3, column 4, some numbers had two decimals and other had three decimals; also in the Table 3, some had no space between a number and left parathesis. f) it seems that the last sentence of the Results is not complete: “Nearly 40% increase in …..”. g) in page 17, “The possible reason may be because patients may fear contracting the pandemic if…” should be “The possible reason may be because patients may fear contracting the novel coronavirus if…”; h) in page 17, “experienced COVID-19 like symptom(s) sedentary lifestyle were found to ...” should be “experienced COVID-19 like symptom(s), and sedentary lifestyle were found to …”

6. PLOS authors have the option to publish the peer review history of their article (what does this mean?). If published, this will include your full peer review and any attached files.

Reviewer #1: No

Reviewer #2: No

---

## [Author Response · Author response to Decision Letter 0]

23 May 2022

May 2022

Rebuttal letter

Manuscript ID: 

PONE-D-21-28858

Title: Effect of COVID-19 pandemic on missed medical appointments among adults with chronic diseases conditions in Northwest, Ethiopia

Tadesse Awoke Ayele, Tesfa Sewunet Alamneh, Habtewold Shibru, MaledeMequanent Sisay, Tesfahun MeleseYilma, MelkituFentie Melak, Telake Azale Bisetegn, Tariku Belachew, Mahteme Haile, Taye Zeru,Mezgebu SelamsewAsres, and Kegnie Shitu

Dear Editor and reviewer, 

We would like to thank for your consideration and suggestion for the betterment of our manuscript and make it more informative. We tried to amend the format of the manuscript according to the journal guidelines and address the questions raised by reviewer on the manuscript. The authors revised the overall manuscript regarding to language usage and grammar errors. In addition, we also consult language experts in our university and amendments were done based on their comments. Our point-by-point responses for each comment and questions are described in detail on the following pages. Further, the details of changes were shown by track changes in the supplementary document attached. 

Editor’s comment

1. Please ensure that your manuscript meets PLOS ONE's style requirements, including those for file naming. The PLOS ONE style templates can be found athttps://journals.plos.org/plosone/s/file?id=wjVg/PLOSOne_formatting_sample_main_body.pdf andhttps://journals.plos.org/plosone/s/file?id=ba62/PLOSOne_formatting_sample_title_authors_affiliations.pdf

Authors’ response: Thank you dear editor for your concern. We tried to adjust the format according to the journal requirements

2. You indicated that you had ethical approval for your study. In your Methods section, please ensure you have also stated whether you obtained consent from parents or guardians of the minors included in the study or whether the research ethics committee or IRB specifically waived the need for their consent..

Authors’ response: Thank you dear editor for your concern. The study participants were individuals who are aged 18 years and above. Therefore, they can give informed consents by themselves and we don’t include other minority groups.

Authors’ response: Thank you dear editor for your concern. We include it as supporting information.

Authors’ response: Thank you dear editor for your concern. We have presented the appropriate data availability statement on the online submission and upload de-identified dataset.

Authors’ response: Thank you dear editor for your concern. We have put it at the end of methods section as per your recommendation. 

To reviewer 1

Reviewer comments 

 Abstract:

1. Results: There seems to be a missing word where it reads ‘most (55.4%) were female.’ Suggest need to add in ‘participants’ or similar.

Authors’ response: Thank you dear reviewer for your concern. Yes, we can’t say most instead of more than half of the study participants. To make the abstract short and precise we have excluded this statement from the abstract because interested readers can catch it from the result section 

2. Conclusion: This is repetitive as it duplicates the reasons of missed appointments which are also found in results. The conclusions would be better to include more about the implications of the findings or the next steps for future research. This relates to a later comment about the conclusion in the main body of the paper.

Authors’ response: Thank you dear reviewer for your concern. We include the implications of the findings in the revised documents.

Introduction:

3. The first paragraph of the introduction would benefit from more references. If the references that are provided also cover earlier points, it might be worth adding the reference number in again. Some points seem to be unreferenced so this would make it clearer. One example of this is ‘Access to health facilities was restricted due to fear of the spread of the virus from patient and health professional perspectives. Patients were scared to go to a health facility to receive their care and treatments.’ This occurs throughout the introduction.

Authors’ response: Thank you dear reviewer for your concern. We add revised this paragraph and include additional reference.

4. In the second paragraph of the introduction you state that ‘Although the virus infects every individual’ this would read better as ‘potentially infects every individual’ as much is still unknown and not everyone is infected.

Authors’ response: Thank you dear reviewer for your concern. We have edited it as per your comments.

5. When you say ‘An estimated 41% of adults had delayed or avoided medical care including urgent or emergency care (12%) and routine care (32%) because of concerns about COVID-19 in the US’ it would be helpful to define the timeframe that this covers.

Authors’ response: Thank you dear reviewer for your concern. We have included the time frame.

6. It would be useful to include a statement on why looking at missed appointments is needed/ relevant/ interesting. What is the impact of missed appointments on health services and patients? (cost and time for health care professionals and unmet need/ delayed treatment for patients).

Authors’ response: Thank you dear reviewer for your concern. We have included an statement entertain what you raised in the last paragraph of the introduction in the revised document.

Methods:

7. Study design and period: ‘Two years retrospective data’ could you just briefly state what type of data this was- e.g. was it electronic patient records etc?

Authors’ response: Thank you dear reviewer for your concern. We used two year data that were extracted from chart and some data were collected by the interview at the time of data collection 

8. Study setting: Of the 9 hospitals that participated in the study, what were their demographics? (e.g. were they rural/ urban etc).

Authors’ response: Thank you dear reviewer for your concern. They are located at urban areas and there deference is level of service they provide but they serve peoples from both urban and rural areas. 

Data collection methods and measurement:

9. This sentence would be worth rewording to make it clearer: ‘During data collection, it has been about a year since the emergence of the COVID-19 pandemic.’ Something like ‘Data collection was conducted approximately a year into the pandemic.’

Authors’ response: Thank you dear reviewer for your concern. We rewrite it in the main document. 

10. On page 7 you mention patient interviews. This is the first mention, and it is not clear what this relates to. Is this a reference to the surveys that were used? It implies that there is a qualitative component to the study, so suggest rewording it.

Authors’ response: Thank you dear reviewer for your concern. We used both primary and secondary data source to answer our research questions. We extract a two year data from chart but there are some variables that were not recorded at the chart. Then we conduct patient interview to collect data for variables that are not found in patient chart. 

Results:

11. Socio-demographic characteristics: In line with earlier comment, when you say ‘A total of 1,833 individuals with common chronic diseases were interviewed’ do you mean ‘A total of 1,833 individuals with common chronic diseases completed the survey.’ Possibly reword to make it clearer they completed the survey.

Authors’ response: Thank you dear reviewer for your concern. We have edited it in the revised version.

12. ‘More than one-fourth (28.0%) of the individuals with a common chronic condition was not able to read and write’ should say ‘were’ not ‘was.’

Authors’ response: Thank you dear reviewer for your concern. We have corrected it in the revised document.

13. There are a few examples in the results sections where the language would benefit from being revised to make it clearer. E.g. ‘About one in four patients (23.63%) have been following for more than 10 years,’ (have been receiving follow-up?) and ‘However, only 33 (1.8%) diagnoses for COVID-19 and all of them were positive,’ (Diagnosed implies that they were positive) ‘Nearly one-third (33.35%) of the patients had identified co-morbid diseases in which hypertension was the commonest type of co-morbidity observed in most of the patients.’ (Duplication- commonest and most- suggest just use one or the other).

Authors’ response: Thank you dear reviewer for your concern. We have checked the overall manuscript regarding language usage and grammar errors as and we try to amend it. In addition, we also consult language experts in our university and amendments were done based on their comments.

14. Missed appointments: Again a few suggestions on improving the language. E.g. ‘About 225 (12.47% (95% CI: 11.13%, 14.20%)) of patients with common chronic diseases have missed their appointment during their follow-up before COVID 19 pandemic’ (remove ‘have’).

Authors’ response: Thank you dear reviewer for your concern. We have removed it

Discussion:

15. You start by discussing generally about Covid and low-income countries, but then move on to talk specifically about the country being studied. ‘There have been different strategies and non-pharmaceutical interventions implemented, however, the implementation level has been very limited due to the cultural and traditional practices in the country’. It would be good to state here when you move on to talk about one specific country to make it clear.

Authors’ response: Thank you dear reviewer for your concern. We have modified it as per your comment.

16. Again a few missed words in this section ‘the rate of COVID-19 transmission is higher United States, Singapore, Canada and Italy.’ Missing ‘in’.

Authors’ response: Thank you dear reviewer for your concern. We include it.

17. Strengths and limitations: There is limited discussion of the strengths and limitations for this study. You mention a few strengths but no discussion of why they are strengths or how they strengthen the study.

Authors’ response: Thank you dear reviewer for your concern. We revise the strength and limitations of the study.

18. I also suggest you could include a section on implications and discuss how this research can help understanding of this topic, implementation of policy or impact on practice. I also think that it would be useful to include any future research needed or [planned on the topic. This also needs to be included in the abstract.

Authors’ response: Thank you dear reviewer for your concern. We include the implication of the study in the both abstract and conclusion section.

Reviewer 2: 

Reviewer comments

1. The authors conducted a retrospective cohort study to evaluate the impact of the COVID-19 pandemic on missed appointments among those with chronic conditions in Northwest of Ethiopia. It was important to know the scale of the impact and the risk factors that were associated with increased risk of no-show. Here are my concerns: I do not know why the author called their study “chart review and cross-sectional survey”. Their data source was electronic chart. I also did not see any conventional survey was conducted. Suggest remove “chart review and cross-sectional survey” and replace it with “a retrospective cohort study”.

Authors’ response: Thank you dear reviewer for your concern. The data source for this study was primary data which was collected cross-sectionally by interview at the time of their hospital visit and secondary data was extracted from patient chart. From the patient chart, we extract 2 years data one year before the COVID-19 pandemic and a year after the pandemic but we only assess two years data retrospectively but we didn’t have control groups. Therefore, retrospective cohort study might not be appropriate for this scenario because we didn’t have control group.

2. Appointment data were extracted during the COVID-19 pandemic and one year before the pandemic for the same patients. Thus, the outcome, missed appointments from these two periods were not independent. However, the authors used an independent t-test to test the difference in missed appointments before and during the pandemic. In addition, t-test may not be the right test for the number of missed appointments. A binomial test is preferred for count data, or just run the negative binomial regression model without including covariates.

Authors’ response: Thank you dear reviewer for your concern. The t-test was conduced to assess the proportion of missed appointment before and during the pandemic was significant or not while the number of missed appointment was used as outcome variable when we assess the determinants of missed appointment. In our case, the observations are not independent and then it is possible to consider paired t-test to determine the difference in proportion of missed appointments before and during the pandemic. However, we only estimate the proportion missed appointment before and during the pandemic with its 95%CI.

3. In the Results, the authors described how they landed on negative binomial regression because of over-dispersion. I recommend a) In Methods, make clear that number of missed appointment is the dependent variable in the negative binomial regression; b) consider the dependence of observations in the regression; c) consider including number of scheduled appointments as an offset in the regression model (natural log(number of scheduled appointments)); when the offset is used, they should remove “The more frequent the visits, the higher the incidence of missed appointments” in page 14, and should not include duration of follow-up because number of scheduled appointments already contained all the information needed.

Authors’ response: Thank you dear reviewer for your concern. We revise according to your comments. Yes, it is better to consider offset variable in count data analysis. Despite the number of missed appointment, duration of follow up is an offset variable for us Because an individuals with longer duration of variable had higher risk of missed appoint and vice versa and. Regarding to the independent assumption, we only use the frequency of missed appointment as dependant variable and consider missed appointment history before he pandemic as predictor. Therefore, independent assumption is not an issue for us. 

4. It was not clear what criteria were used to determine that negative binomial regression model was the best. Please provide a reference in page 14.

Authors’ response: Thank you dear reviewer for your concern. The commonly used count data analysis is Poisson distribution with constrained assumption which is the mean and variance should be equal. Because of the violation of this assumption, we consider negative binomial regression.

5. These sentences in Discussion are confusing: “Still, after a year, the pandemic remains very challenging, especially for low-income countries. There have been different strategies and non-pharmaceutical interventions implemented, however, the implementation level has been very limited due to the cultural and traditional practices in the country.” We are in the third year of the pandemic. I do not know what “strategies” and “interventions” were referred to and for what purposes. Suggest removing these sentences.

Authors’ response: Thank you dear reviewer for your concern. Yes we are in the third year but this study was based on the data a year after the declaration of pandemic. We have revised it in the main document.

6. The author did not distinguish missed appointments, cancelled appointments, and rescheduled appointments. Were cancelled and rescheduled appointments counted as missed appointments? If they were, how many of those missed appointments were cancelled or rescheduled? Please discuss this issue.

Authors’ response: Thank you dear reviewer for your concern. We consider both cancelled and rescheduled appointments counted as missed appointments. But we consider it as our limitation because considering rescheduled appointments as missed appointments might overestimate the number missed appointments.

7. There were many grammar-like errors in the manuscript, some examples are: a) lower case for “No-show rate” in page 5; b) there should be a space between 1005 and (55.37%) in page 8; c) throughout the manuscript, insert a space in 95%CI (like 95% CI); d) need define abbreviations before using them (e.g., CIRR, AIRR); e) decimals were not consistence, for example, in Table 3, column 4, some numbers had two decimals and other had three decimals; also in the Table 3, some had no space between a number and left parathesis. f) it seems that the last sentence of the Results is not complete: “Nearly 40% increase in …..”. g) in page 17, “The possible reason may be because patients may fear contracting the pandemic if…” should be “The possible reason may be because patients may fear contracting the novel coronavirus if…”; h) in page 17, “experienced COVID-19 like symptom(s) sedentary lifestyle were found to ...” should be “experienced COVID-19 like symptom(s), and sedentary lifestyle were found to …”

Authors’ response: Thank you dear reviewer for your concern. We tried to revise the overall manuscript regarding to any typological and grammar errors.

---

## [Editor Report · Decision Letter 1]

22 Jun 2022

PONE-D-21-28858R1Effect of COVID-19 pandemic on missed medical appointments among adults with chronic diseases conditions in Northwest, EthiopiaPLOS ONE

Dear Dr. Shitu,

Thank you for submitting your manuscript to PLOS ONE. After careful consideration, we feel that it has merit but does not fully meet PLOS ONE’s publication criteria as it currently stands. Therefore, we invite you to submit a revised version of the manuscript that addresses the points raised during the review process.

ACADEMIC EDITOR:

Appreciate that a conclusion was added, but it was too long, suggest the following:

The rate of missed appointment increased significantly during the COVID-19 pandemic. Older age, longer duration of follow up, more frequent follow-up, out-of-pocket expenditure for health service, history of poor follow-up, and sedentary lifestyle had positive relationship with missed appointments during the pandemic. Therefore, it is important to give special emphasis to individuals with these risk factors while designing and implementing policies and strategies for peoples with chronic diseases to ensure the continuity of care and to avoid the long-term impact on their health.

Need add the following before the last sentence of "Data Collection Methods and Measurements"

We extracted two-year’s data from medical chart (the primary data source). For those variables that were not recorded in the chart, we conducted patient interview to collect the data (the secondary data source).

It is good to see reasons for strengths and limitations. But there are some language issues. Suggest changing

“The other limitations include retrospective nature of the study might underestimate the outcome variable. Because when the date at for an appointments were not documented, we could not assured weather an individual missed the appointment or not. Moreover, this study considers both cancelled and rescheduled appointments as missed appointments. Though considering rescheduled appointments as missed appointments might overestimate the number missed appointments.”

to

“Another limitation is the retrospective nature of the study. When the date for an appointment was not documented, we could not determine whether an individual actually missed the appointment or not. Moreover, this study considered both cancelled and rescheduled appointments as missed appointments and might have overestimated the number of missed appointments.”

We look forward to receiving your revised manuscript.

Kind regards,

Stanley Xu

Guest Editor

PLOS ONE

Journal Requirements:

Additional Editor Comments:

thank you for addressing the comments of reviewers and the paper improved significantly. As one of the reviewers and now a guest editor, I have some remaining minor issues.
---

## [Author Response · Author response to Decision Letter 1]

19 Aug 2022

August, 2022

Rebuttal letter

Manuscript ID: PONE-D-21-28858R1 

Effect of COVID-19 Pandemic on Missed Medical Appointment among Adults with Chronic Disease Conditions in Northwest Ethiopia

Tadesse Awoke Ayele, Tesfa Sewunet Alamneh, Habtewold Shibru, Malede Mequanent Sisay, Tesfahun MeleseYilma, MelkituFentie Melak, Telake Azale Bisetegn, Tariku Belachew, Mahteme Haile, Taye Zeru,Mezgebu SelamsewAsres, and Kegnie Shitu

Dear Editor,

We would like to thank for your consideration and suggestion for the betterment of our manuscript and make it more informative. We tried to amend the format of the manuscript according to the journal guidelines and address the questions/suggestions forwarded by the editor. Our point-by-point responses for each comment and questions are described in detail on the following pages. Further, the details of changes were shown by track changes in the supplementary document attached to the online submission system. Thanks.

Editor’s comment

Appreciate that a conclusion was added, but it was too long, suggest the following:

The rate of missed appointment increased significantly during the COVID-19 pandemic. Older age, longer duration of follow up, more frequent follow-up, out-of-pocket expenditure for health service, history of poor follow-up, and sedentary lifestyle had positive relationship with missed appointments during the pandemic. Therefore, it is important to give special emphasis to individuals with these risk factors while designing and implementing policies and strategies for peoples with chronic diseases to ensure the continuity of care and to avoid the long-term impact on their health.

Response: Thank you so much for the comment and suggestions: The revision has been made accordingly (See the conclusion in the abstract of the revised manuscript).

Need add the following before the last sentence of "Data Collection Methods and Measurements"

We extracted two-year’s data from medical chart (the primary data source). For those variables that were not recorded in the chart, we conducted patient interview to collect the data (the secondary data source).

Response: Thank you. Changes have been made accordingly (See the revised manuscript line 146-147 on page 7 of the revised manuscript)

It is good to see reasons for strengths and limitations. But there are some language issues. Suggest changing

“The other limitations include retrospective nature of the study might underestimate the outcome variable. Because when the date at for an appointments were not documented, we could not assured weather an individual missed the appointment or not. Moreover, this study considers both cancelled and rescheduled appointments as missed appointments. Though considering rescheduled appointments as missed appointments might overestimate the number missed appointments.”

to

“Another limitation is the retrospective nature of the study. When the date for an appointment was not documented, we could not determine whether an individual actually missed the appointment or not. Moreover, this study considered both cancelled and rescheduled appointments as missed appointments and might have overestimated the number of missed appointments.”

Response: Revision has been made accordingly and your suggestions were super helpful (See the revised manuscript line 379 to 383 on page 21 of the revised manuscripts).

Journal Requirements:

Response: References have been double checked. We couldn’t not any retracted article in our reference list. However, we have made some revisions on some of the references involving adding URL address for internet/website references (See the reference section of the revised manuscript with track changes).

---

## [Editor Report · Decision Letter 2]

24 Aug 2022

Effect of COVID-19 Pandemic on Missed Medical Appointment among Adults with Chronic Disease Conditions in Northwest Ethiopia

PONE-D-21-28858R2

Dear Dr. Shitu,

We’re pleased to inform you that your manuscript has been judged scientifically suitable for publication and will be formally accepted for publication once it meets all outstanding technical requirements.

Kind regards,

Stanley Xu

Guest Editor

PLOS ONE
---

## [Editor Report · Acceptance letter]

22 Sep 2022

PONE-D-21-28858R2 

Effect of COVID-19 Pandemic on Missed Medical Appointment among Adults with Chronic Disease Conditions in Northwest Ethiopia 

Dear Dr. Shitu:

I'm pleased to inform you that your manuscript has been deemed suitable for publication in PLOS ONE. Congratulations! Your manuscript is now with our production department. 

Kind regards, 

on behalf of

Dr. Stanley Xu 

Guest Editor

PLOS ONE